# Electrospun Nanofibers Loaded with *Plantago major* L. Extract for Potential Use in Cutaneous Wound Healing

**DOI:** 10.3390/pharmaceutics15041047

**Published:** 2023-03-24

**Authors:** Javier M. Anaya-Mancipe, Vanessa M. Queiroz, Rafael F. dos Santos, Rosane N. Castro, Verônica S. Cardoso, Alane B. Vermelho, Marcos L. Dias, Rossana M. S. M. Thiré

**Affiliations:** 1COPPE/Program of Metallurgical and Materials Engineering—PEMM, Universidade Federal de Rio de Janeiro—UFRJ, Rio de Janeiro 21941-599, RJ, Brazil; javier.anaya@metalmat.ufrj.br (J.M.A.-M.);; 2Institute of Macromolecules Professor Eloisa Mano—IMA, Universidade Federal do Rio de Janeiro—UFRJ, Rio de Janeiro 21941-598, RJ, Brazil; 3Chemistry Institute, Universidade Federal Rural do Rio de Janeiro—UFRRJ, Seropédica 23890-000, RJ, Brazil; 4Bioinovar—Instituto de Microbiologia Paulo de Góes, Universidade Federal do Rio de Janeiro—UFRJ, Rio de Janeiro 21941-902, RJ, Brazil

**Keywords:** electrospinning, nanofibers, polycaprolactone, *Plantago major* L., drug delivery, wound dressing

## Abstract

*Plantago major* L. is a plant available worldwide that has been traditionally used for several medical applications due to its wound healing, anti-inflammatory, and antimicrobial properties. This work aimed to develop and evaluate a nanostructured PCL electrospun dressing with *P. major* extract encapsulated in nanofibers for applications in wound healing. The extract from leaves was obtained by extraction in a mixture of water:ethanol = 1:1. The freeze-dried extract presented a minimum inhibitory concentration (MIC) for *Staphylococcus Aureus susceptible and* resistant to methicillin of 5.3 mg/mL, a high antioxidant capacity, but a low content of total flavonoids. Electrospun mats without defects were successfully produced using two *P. major* extract concentrations based on the MIC value. The extract incorporation in PCL nanofibers was confirmed using FTIR and contact angle measurements. The PCL/*P. major* extract was evaluated using DSC and TGA demonstrating that the incorporation of the extract decreases the thermal stability of the mats as well as the degree of crystallinity of PCL-based fibers. The *P. major* extract incorporation on electrospun mats produced a significant swelling degree (more than 400%) and increased the capacity of adsorbing wound exudates and moisture, important characteristics for skin healing. The extract-controlled release evaluated using in vitro study in PBS (pH, 7.4) shows that the *P. major* extract delivery from the mats occurs in the first 24 h, demonstrating their potential capacity to be used in wound healing.

## 1. Introduction

The skin is one of the body’s major organs, as it plays the role of maintaining internal fluid homeostasis. It provides a barrier against environmental, chemical, mechanical, and other damages. This is why skin damage can be considered a major health problem [1,2,3]. Although the skin has the capacity for self-healing, wound healing can be affected and/or delayed due to infection and necessitate the application of antibiotics [4].

Advances in regenerative medicine related to wound treatment have led to the production of new nanostructured dressings that, in addition to serving as a protective barrier for the wound, maintain effective conditions to promote cell proliferation and migration [5,6]. Implementing intelligent and bioactive dressings, which have structures that mimic the extracellular matrix of the skin (ECM), and maintain a moist environment, while preventing bacterial colonization in the wound bed through the controlled release of drugs/antibiotics, active ingredients, and/or factors that accelerate the regenerative processes, favoring the repair of damaged tissue [7,8,9].

In recent years, it has become popular to include traditional medicine, in which drugs of natural origin, such as waxes, extracts, dyes, and essential oils obtained from leaves, seeds, flowers, etc., are widely used combined with encapsulation methodologies such as nanoparticles, gels, and nanofibers that together have shown promising results for its application in clinical practice [10,11,12,13,14,15]. One of the techniques that have been gaining prominence in this application is electrospinning, due to its wide variety of materials and solvents used that allow the production of nanostructured biomedical materials.

Electrospinning is a promising technique to create nanofibrous mats with efficient morphological and mechanical properties, allowing those materials to have a good capacity for promoting the recovery of structurally damaged tissues close to the extracellular matrix (ECM) of the skin [16,17]. These structural features have called attention to their application in regenerative medicine, which uses natural and synthetic polymers with the ability to encapsulate various active materials, drugs, and/or growth factors [18]. One of the main advantages of this process is its versatility and ease of production, producing nanostructured matrices with an ECM design that allows for rapid cell growth and adhesion. In addition, electrospinning is an easy way to functionalize the nanofibers, incorporating a wide range of bioactive additives for wound healing [19].

In the last years, advancements in electrospinning technique for skin wound healing have been reported in the literature. Li et al. (2022) studied the production of fibrous matrices of GSNO-grafted Trp-PEUs/PCL mats for the first time to serve as a prolonged NO delivery up to 408 h after grafted with GSNO. The mats promote the growth of cells, regulate inflammatory factors, and improve collagen deposition in the wound, eventually accelerating wound healing. In conclusion, this paper showed a new idea that promoted the NO-release wound dressing to treat infection in wounds, presenting efficient results for biofilm treatment. Traditional antimicrobial treatments, such as macrolides, tetracyclines, and quinolones, have been proven to be effective against bacterial infections, but they perform poorly in annihilating biofilms [20]. In 2023, Qi et al. studied strategies for producing PCL and silk fibroin (SF)-based nanostructured mats with different yarns for tissue engineering applications. The authors observed that the inclusion of SF in PCL mats generated improvements in the wettability of the mats as well as better cellular response with reduced inflammatory response, thus demonstrating the potential of these biomaterials for application as a nanofibrous biotextile in tissue engineering. In general, the authors developed a simple, quick, and cost-effective method by combining conventional electrospinning process with hand winding and stretching post-treatment as a nanofiber-constructed yarns (NYs) with great potential for tissue engineering and regenerative medicine [21].

Polycaprolactone (PCL) is one of the most used polymers in electrospinning. It is a biodegradable polyester, highly used in the production of medical devices due to its good biocompatibility and, in addition, offers excellent characteristics for its use as an encapsulating material for the controlled release of drugs and/or active material, making it an effective polymer for biomedical applications [22,23]. This is due to its elastomeric characteristic at room temperature, giving excellent mechanical properties for dermal applications. The greater movement of chains at room temperature attributed to its low glass transition temperature (Tg = −60 °C) makes it an excellent encapsulating agent for drug and/or active materials for their subsequent use in drug delivery. Compared to the other biopolymers such as PLA, PLGA, etc., PCL also presents stability in processing and storage as compared to other polymers. On the other hand, one of the main disadvantages of PCL is its low degradation rate recorded in the literature of approximately 2 years. However, Dias and colleagues (2022) conducted in vitro and in vivo degradation studies of electrospun PCL nanofibers, finding in vitro degradation times of 90 days, which may be attributed to their high surface area [24]. 

*Plantago major* L. (*P. major*) is a perennial plant that belongs to the Plantaginaceae family. Originating from Europe, it is the most widely used species in both traditional and modern medicine [25,26]. Its use is mostly related to the implementation of the leaves in the preparation of extracts, essential oil, and pastes, among others, with excellent results for the treatment of cutaneous wounds. The activity is mainly attributed to its phytochemical components such as polysaccharides, lipids, derivatives of caffeic acid, flavonoids, iridoid glycosides, and terpenoids, offering anti-inflammatory, antioxidant, analgesic, immunomodulatory, anti-ulcer, and antibiotic capacities. These pharmacological properties make it a promising phytochemical for biomedical and regenerative applications [27,28,29,30,31,32].

The wound-healing capacity of the leaf extract has been proved by “in vivo” tests with animals and humans, using different pharmacology forms, such as ointment [33] and topical gel [34]. *P. major* is a medicinal plant widely used in traditional medicine worldwide as a phytopharmaceutical for the treatment of injuries, with anti-inflammatories, antifungals, antiseptics, and analgesic properties. One of its main advantages lies in its easy availability, as even though it is originally from Europe, it has spread throughout the world, favoring and/or facilitating its availability in most countries, a fact that has popularized its use, making it a promising alternative as a phytopharmaceutical. On the other hand, as demonstrated in the literature, the *P. major* extract effectively accelerated the healing of dorsal cervical injuries in mice, with minimal adverse reaction compared to traditional treatment for burns such as silver sulfadiazine, indicating good reepithelization, low inflammatory cell concentration, and good granulation tissue formation [33,35,36,37].

Amini and collaborators (2010) evaluated the regenerative power of the hydroalcoholic extract of *P. major* L. in burns [37]. In vivo tests were performed in rats by varying the concentration of the extract applied at 7, 14, and 21 days. The histological analyses demonstrated a significant variation in the group of animals that received the extract with a concentration of 50% *w/v* producing therapeutic action from day 21 of the study. Reina et al. (2013) evaluated in vitro effects in human neutrophils of *baicalein* and *aucubin* (flavonoid and glycoside, respectively) present in *P. major* leaf extract showing no cytotoxicity in LHD assays as well as good antioxidant properties of baicalein for its use in dressings [38].

However, few works reporting studies related to the encapsulation of *Plantago major* in electrospun nanofibers were found in the literature. The search was performed using these keywords: *Plantago major*, nanofibers, and electrospinning. With these keywords, only two studies were found. The first one was performed in 2019 by Golkar et al. [39], who developed electrospun nanofibers with average diameters smaller than 250 nm using a blend of *P. major* seed mucilage (MSM) and poly(vinyl alcohol)—PVA in equal mass ratio (1:1). The authors observed a significant increase in cell proliferation when MSM was incorporated, demonstrating its ability to be implemented as a platform for cell culture and/or drug delivery.

Recently, in 2022, de Castro et al. [40] developed electrospun nanofibers of hyaluronic acid (HA) and PVA loaded with commercial aqueous extract from leaves of *Plantago major* for application in smart dressings. In this work, the authors evaluated the spinnability of the blend using maleic acid as a crosslinking agent, as well as the water adsorption capacity and the interaction of the extract with the polymer blend.

Thus, the main objective of this work was to produce nanostructured bioactive polycaprolactone mats via monolithic electrospinning with the incorporation of a phyto-drug obtained from *Plantago major* L. leaves aiming at potential application in skin tissue regeneration. For this, the chemical and pharmacological composition of the *P. major* extract was evaluated using HPLC. The electrospun mats were characterized morphologically using SEM and the encapsulation of the extract was evaluated by using FTIR. The interaction of the extract and PCL was evaluated using DSC, TGA, and contact angle. Finally, its application as a dressing was evaluated in vitro considering the exudate adsorption by swelling assay and drug delivery capacity.

## 2. Materials and Methods

### 2.1. Materials

Polycaprolactone—PCL in pellets (Mw: 70,000–90,000 g/mol; GH: 98%) was purchased from Sigma-Aldrich, São Paulo, Brazil. Dry *Plantago major* L. (*P. Major*) leaves were acquired from RioFlora (Ervas Medicinais), Rio de Janeiro, Brazil. As solvents, we used dichloromethane (DCM), *N,N*-dimethylformamide (DMF), and ethanol—PA acquired from Vetec Quimica Fina (Rio de Janeiro Brazil). Spectroscopy-grade methanol was purchased from Vetec (Rio de Janeiro, Brazil). DPPH (2,2-diphenyl-1-picryl-hydrazine) and Folin–Ciocalteau reagent were acquired from Sigma-Aldrich Chemie (Steinheim, Germany). Sodium phosphate dibasic and potassium phosphate monobasic for phosphate buffer solution (PBS) preparation were purchased from Vetec Química Fina (Rio de Janeiro, Brazil).

### 2.2. Plantago major L. Extract Preparation

The *Plantago major* L. hydroalcoholic extract was prepared following the method described by Mello et al. (2015) with some modifications [41]. Dried *P. major* leaves (10 g) were weighed and crushed using a mortar. The leaves were then placed in a flask with a hydroalcoholic solution (1:1) of ethanol/MiliQ water for 24 h at room temperature. Afterward, the suspension was put in an ultrasonic bath for 1 h, followed by vacuum filtration and roto-evaporation at 50 °C to decrease its volume by 50%. The *P. major* extract was cooled down to room temperature, and then frozen and freeze-dried for 48 h to remove the moisture completely. Finally, the dry extract was stocked to ~4 °C.

### 2.3. Polymeric Solutions

Three PCL solutions were prepared adding different mass concentration of the *Plantago major* dry extract. A solvent system composed by DCM/DMF (7:3) (10 wt/v% PCL) was used [42]. For this, 0.5 g of PCL pellets were weighed and solubilized in 3.5 mL dichloromethane, while the freeze-dried extract was suspended in 1.5 mL of DMF at two concentrations (26.5 mg and 53.0 mg) under magnetic stirring and room temperature. Subsequently, the two solutions (PCL/DCM and *P. major*/DMF) were mixed under magnetic stirring for another half hour or until a homogeneous mixture was obtained. The solutions were named as PCL_0.0, PCL_5.3, and PCL_10.6 according to the content of dry extract in the final solution. These contents corresponded to the values of the minimum inhibitory concentration (MIC) and twice MIC, respectively, of the dry extract as determined following the procedure described in Section 2.7.1.

### 2.4. Electrospinning Process

Electrospun PCL mats with and without *Plantago major* L. extract were prepared using a horizontal electrospinning apparatus composed of Glassman High Voltage source (USA) model PS/FC 60p02.0-1, a KDS 100 series syringe pump, plastic syringe with a metallic needle, and a grounded aluminum plate collector. To facilitate samples remotion, aluminum foils were used over the plate collector. Five mL of each solution were transferred on a plastic syringe and metallic needle with 0.5 mm of internal diameter (22 gauds). The flow rate was maintained at 0.75 mL/h, while the voltage applied was varied in the range of 15 to 20 kV, and the working distance was 12 cm. The obtained mats were named as PCL_0.0, PCL_5.3, and PCL_10.6 mats. The spinnability of the solution was evaluated by using morphological analyses. To verify the effect of adding extract on the fiber diameter, an analysis of variance (ANOVA) post hoc test–Fisher’s least significant difference (LSD) was performed adopting a 0.05 significance level (*p* < 0.05) [43].

### 2.5. Fhysico-Chemical Characterization

#### 2.5.1. Determination of Phenolic Compounds in *P. major* Extract

The *Plantago major* extract solutions were prepared by solubilization of 1.0 mg/mL in ethanol/water (7:3). Subsequently, the solutions were used to calculate the concentration of total phenols and flavonoids, following the methods described by Souza et al. [44].

The total phenolic contents of the extracts of *P. major* were assayed using the Folin–Ciocalteu method as described earlier [45]. The quantity of total phenols was calculated using an aliquot of *P. major* solution (50 µL) and mixed with 2.5 mL of Folin–Ciocalteu reagent (1:10) and 2.0 mL of Na_2_CO_3_ (4%). The solution was incubated for 5 min at 50 °C. Subsequently, measurements were performed using UV-vis spectroscopy using MiliQ water as blank at 760 nm. A standard curve was obtained for gallic acid, and the phenolic content was determined from the extrapolation of this curve [45]. The measurements were carried out in triplicate.

The total flavonoid contents of the extracts of *P. major* were determined by using the aluminum chloride colorimetric method as described by Salgueiro et al. [46]. Briefly, an aliquot of 400 µL of the extract was placed in a 10 mL volumetric flask and mixed with 200 µL of methanolic solution with AlCl_3_ (2.0 wt. %), and completed volume using spectroscopy-grade methanol to 10 mL. The mixture was left in the dark for 30 min at room temperature, and the absorbance of the reaction mixture was recorded using a spectrophotometer at 425 nm. The measurements were made in triplicate.

The antioxidant capacity of the *Plantago major* extract was performed using the radical scavenging method (DPPH) [41]. Aliquots of 70 µL of the extract were mixed with 29 µL of 0.3 mmol/L of fresh DPPH solution and solubilized in methanol (spectroscopy grade). The final solution was incubated at room temperature and left in the dark for 30 min. The absorbance of the reaction mixture was recorded at 517 nm after this time using an ELISA plate reader (Bio-Rad, São Paulo, Brazil).

High-performance liquid chromatography with diode arrangement detector (HPLC/DAD) was performed with a Shimadzu Prominence Auto Sampler (SIL-20A) HPLC system (Shimadzu, Kyoto, Japan), equipped with Shimadzu LC-20AT reciprocating pumps connected to a DGU 20A5 degasser with a CBM 20A integrator, SPD-M20A diode array detector, and an LC solution software. Reverse phase chromatographic analyses were carried out under gradient conditions using a C18 column (4.6 mm × 250 mm, 5 μm diameter particles, Betasil, Thermo Scientific, Waltham, MA, USA) held at 40 °C. The mobile phase consisted of Milliq water containing 1% acetic acid (A) and methanol (B), the flow rate was 1.0 mL/min, the injection volume was 15 μL, and the analysis wavelength was 330 nm. Sample extracts were analyzed at a concentration of 5 mg/mL. The gradient program was started with 15% of B for 3 min and was then changed to obtain 60% B in 20.00 min, and then 80% B in 25 min. The phenolic compounds contained in extracts were recognized by comparing retention time and UV absorption spectra with those of the commercial standards.

#### 2.5.2. Viscometry

Rheological measurements of PCL_0.0, PCL_5.3, and PCL_10.6 solutions were performed on ViscoQC-100, Rotational viscosimeter (Anton Paar Trading Co., Shanghai, China) using the CC18 spindle varying the shear rate at room temperature.

#### 2.5.3. Scanning Electron Microscopy—SEM

The morphological evaluation of electrospun nanofibers with and without *Plantago major L.* extract was carried out with SEM using a Tescan Vega3 (Brno, Czech Republic) with an acceleration of 10 kV. All samples were gold-coated prior to the analysis using sputter equipment Denton Vacuum—Desk V (Moorestown, NJ, USA) for 120 s at 30 mA. The samples were processed using the Size Meter 1.1 software for obtaining the average diameters of fibers (50 measurements for each sample—n = 3).

#### 2.5.4. Wettability Assay

The contact angle analysis was used to evaluate the wettability of the PCL/*P. major* mats with the variation of hydroalcoholic extract (P. major) encapsulated, using a goniometer Ramé-Hart NRL A 100-00. A drop of distilled water (~4 µL) was deposited on the surface of each sample evaluated at room temperature. Contact angles were measured in triplicates.

#### 2.5.5. Fourier Transformed Infrared Spectroscopy—FTIR

The incorporation of the *P. major* extract in electrospun mats was confirmed using a Nicolet Fourier Transform Infrared Spectrometer equipped with an attenuated total reflectance (ATR) accessory (model 6000, Thermo Scientific). The analysis was performed in the region of 4000–650 cm^−1^, with 64 scans and a resolution of 4 cm^−1^.

#### 2.5.6. Thermal Behavior using DSC and TGA/DTA

The thermal behavior of the electrospun mats was evaluated using differential scanning calorimetry (DSC) in Hitachi—DSC 7020 Thermal Analysis system equipment. Each sample (10 mg) was subjected to two heating cycles and one cooling cycle, which were carried out at a rate of 10 °C/min under a nitrogen atmosphere with a flow rate of 50 mL/min. The first heating cycle was conducted from 25 to 90 °C, followed by a cooling cycle to −80 °C and subsequent heating from −80 to 90 °C. The degree of crystallinity of the material (*X_c_*) was calculated by Equation (1).
(1)$$X_{c}=\frac{\Delta H_{f}}{\Delta H^{o}{}_{f}}$$
where Δ*H_f_* is the melting enthalpy of the endothermic peak of the DSC thermogram (second heating), while Δ*H°_f_* = 151.7 J/g is the theoretical melting enthalpy for a 100% crystalline PCL sample [42]. 

The thermal stability and weight loss of the electrospun mats (PCL_0.0, PCL_5.3, PCL_10.6) and pure dry extract were evaluated by using thermogravimetry analysis using Shimadzu TGA-50 equipment with a heating range of 25 °C to 700 °C and a heating rate of 10 °C/min under N_2_ atmosphere.

#### 2.5.7. In Vitro Swelling Degree

The swelling capacity of PCL and PCL/*P. major* electrospun nanofibers was tested in phosphate-buffered saline solution (PBS, pH = 7.4). The swelling degree (*SD*) of the samples was calculated in triplicates (n = 3) using Equation (2).
(2)$$SD\left(\%\right)=100\times \frac{W_{S}-W_{D}}{W_{D}}$$
where *W_S_* is the weight of the swelled sample at a prescribed time and *W_D_* is the initial mass of the sample (dry weight).

#### 2.5.8. Water Vapor Transpiration Rate Assay (WVTR)

The water vapor permeation test was performed following the ASTM D 1653. A permeability cup (Payne cup) filled with 15 mL of ultrapure water was used, and the electrospun mats were placed on its opening inside a closed glass chamber at room temperature (28–32 °C). A digital hygrometer was used to continuously monitor the relative humidity (RH) percentage (40–42%), and a reservoir of sodium pentoxide (C_5_H_11_NaO) was also present. The evaporation of water through the test mats was monitored by measuring the weight loss of the cup.

### 2.6. In Vitro Release of the Extract

#### In Vitro *Plantago major* Extract Release Evaluation

*Plantago major* L. extract delivery test from PCL electrospun mats (2 × 2 cm^2^) was carried out in 50 mL of PBS solution (pH 7.4) at 37 °C (n = 3). Samples were incubated in an orbital shaker (MSM130/B, M.S. Mistura) at 70 RPM for 48 h. Aliquots of 5 mL were collected from the medium at different periods: 0.5, 2, 4, 24, and 48 h. The same volume of fresh PBS solution was added to the systems after aliquots removal. The amount of the extract released was evaluated by using UV-Vis spectrophotometry, using a UV-Vis Thermo Scientific Evolution 600 spectrophotometer at 327 nm. In order to correlate the absorbance values with extract concentrations in the media, the calibration curve of *P. major* dry extract in PBS solution was plotted. Since the addition of fresh solution alters the *P. major* extract concentration, it was corrected by using Equation (3).
(3)$$Correction\;Factor={\left(\frac{V_{0}}{V_{0}-V_{Aliq}}\right)}^{n-1}$$
where *V*_0_ is the initial volume used in the assay (*V*_0_ = 50 mL), *V_Aliq_* is the aliquot used in the test (5 mL), and n is the number of samples used [47].

For the evaluation of extract loading and encapsulation efficiency, each electrospun mat was analyzed using a UV spectrophotometer Shimadzu (UV-2600) at 327 nm, using an accessory for thin films. The calibration curve of *P. major* dry extract in PBS solution was also used to obtain the extract concentration. The drug-loading capacity (*DL*) and encapsulation efficiency (*EE*) were calculated using Equations (4) and (5). The analysis was carried out in triplicate. The results are presented with standard deviation error (SDE) [48].
(4)$$DL=100\times \frac{Amount\;of\;P.\;major\;in\;the\;PCL\;electrospun\;mat}{Amount\;of\;the\;PCL\;electrospun\;mat}\;$$
(5)$$EE=100\times \frac{Amount\;of\;P.\;major\;in\;the\;PCL\;electrospun\;mat}{Amount\;of\;the\;P.\;major\;initial}\;$$

### 2.7. In Vitro Microbiological Testing

#### 2.7.1. In Vitro Minimum Inhibitory Concentration (MIC) Test

The minimum inhibitory concentration (MIC) of the produced *P. major* L. extract was evaluated as the lowest concentration capable of in vitro inhibition against two different strains of *Staphylococcus aureus*, one susceptible (MSSA, ATCC 29213) and another resistant to methicillin (MRSA, ATCC 33591). The MIC was determined following the Performance Standards of Antimicrobial Susceptibility Testing by the Clinical and Laboratory Standards Institute in document M100, according to the method described by Mouro et al. 2021 [45].

#### 2.7.2. Antimicrobial Assay of *P. major*

For antimicrobial analysis of the hydroalcoholic extract, two bacteria were used, one gram-negative *Escherichia coli* (CECT 434) and the other gram-positive *Staphylococcus aureus* (CECT 86). The strains were stored in Tryptone Soy Broth (TSB, Scharlab) with 20% glycerol at −80 °C. The culture broth was maintained in STA-agar solution at 4 °C; following the method used by Calatayud et al. [49] with modifications, each strain was transferred to 10 mL of TSB and incubated at 37 °C for 18 h to obtain cells in initial stationary phase.

Cell cultures were performed in the stationary phase, with an optical density of 0.9 at 625 nm. The bacteria culture was diluted in TSB and incubated at 37 °C under an optical density of 0.2 at 625 nm (10^5^ CFU/mL) in exponential phase 100 µL of Mueller Hilton Brith (MHB, Scharlab). Samples of the mats with approximate mass of 35 mg (1.0 cm in diameter) were placed in each of the tubes and incubated at 37 °C for 18 h. A tube with a PCL_0.0 mat was used as positive control, while the hydroalcoholic extract deposited on filter paper (1.0 cm in diameter) and dry posteriorly for used as negative control (method adapted from CLSI M07-A9) [50].

For the statistical analysis, The R (version 1.4.1106) software was used to perform the analysis Kruskal–Wallis and the means were compared by Fisher’s least significant difference with a 95% confidence level (*p*-value < 0.05).

## 3. Results and Discussion

### 3.1. Composition Evaluation of Plantago major Extract

Flavonoid and phenolic compounds are the main secondary metabolites identified for the application of plants in medicine and pharmaceutical applications. These compounds are considered effective antimicrobial and antioxidant sources. Therefore, the identification of these chemicals is very important when natural products from parts of plants such as seeds, leaves, and others are used in pharmaceutical applications.

Total phenolic (TP) compounds present in the *P. major* freeze-dried extract was determined as 65.4 ± 0.002 mg GAE/g, a value close to those recorded in the literature for ethanolic extract of this plant (65.53 ± 0.034 mg/g). However, this result was twice higher as for the hydroalcoholic extract (32.12 ± 2.75 mg/g), which was the same extraction solvent used in this work [39]. The phenolic compounds can neutralize free radicals and activate anti-oxidant enzymes, among other metabolic activities, etc. [51].

The total flavonoid (TF) content was determined by using the aluminum chloride complexation method which is specific for flavones and flavanols. An analytical curve of quercetin was used as standard (R^2^ = 0.99) for quantification. The TF results for the *P. major* freeze-dried extract were quantified as 5.7 ± 0.001 mg/g (EQ). However, this valor is five times lower than values reported in the literature for flavonoids in ethanolic extract (28.76 ± 1.05 mg/g EQ) and 3.5 times lower (19.93 ± 0.51 mg/g EQ) for the hydroalcoholic extracts [52].

The antioxidant effect of the *P. major* extract was evaluated by DPPH scavenging due to its ability to scavenge DPPH free radicals by hydrogen donation. The activity was expressed in µg/mL and represents the concentration of the extract needed for 50% inhibition of free radicals (IC_50_). On the other hand, it was reported that a lower IC_50_ indicates higher activity in this assay. It was found that the IC_50_ of the hydroalcoholic extract produced in this work is 14.55 µg/mL (r^2^ = 0.99; n = 4), demonstrating a high ability to reduce the stable radical and exhibit effective scavenging activity compared to the antioxidant capacity of the *P. major* extracts extracted with different solvents. Karima et al. (2015) evaluated that the *P. major* extracts obtained with ethyl acetate (Ac) presented strong antioxidant activity (IC_50_ of 12.85 ± 0.27 µg/mL), while those obtained with aqueous extract (Aq) showed low light activity (IC_50_ of 109.67 ± 0.21 µg/mL). Moreover, the extract produced from petroleum ether fraction (PE) presented even lower antioxidant capacity (439.84 ± 6.51 µg/mL). Thus, it is possible to say that the extract produced in this work offers an effective antioxidant power since the IC_50_ is close to the IC_50_ recorded for Ac [53]. The high antioxidant activity may be attributed to the high phenolic content. Extracts with high antioxidant capacity have been shown to facilitate wound healing, since they can remove the products of inflammation in wound bead, i.e., the excess of proteases and the reactive oxygen species (ROS) [54].

Studies reported in the literature demonstrate a direct relationship between the various components of natural extracts used in traditional medicine and their applicability. For example, secondary metabolites such as *squalene* and *docosan*, which are responsible for the anti-inflammatory and antimicrobial activity, respectively, are found in extracts produced from leaves of medicinal plants such as *Plantago major* L. [33,34,55]. Based on this fact, the identification of secondary metabolites in the extract produced in this study is of utmost importance. For this reason, HPLC, coupled with diode array detection, was used to identify the particular phenolic compounds of the freeze-dried hydroalcoholic extract by comparing the retention times of the detected peaks with those of the pure compounds, and the chromatogram is presented in Figure 1. *Plantago major* L. extract is rich in tannins, flavonoids, caffeic acid, and other substances that demonstrate its efficacy to be used on skin wounds, which generates its antimicrobial and antifungal action [29]. The results of the HPLC analyses of the methanolic extracts of the investigated plants were recorded at 330 nm, and some peaks were identified by comparison with standards. From this analysis, four hydroxycinamic acids, 4-O-caffeoylquinic acid, chlorogenic acid, caffeic acid, and rosmarinic acid, and one flavone luteolin were the possible compounds to be detected in the extracts. The rosmarinic acid (tR = 13.830 min) is the most abundant phenolic compounds present in extracts. The bioactivities of *Plantago major* L. are attributed to these chemical constituents and the various bioactive compounds present in the methanolic extract of this plant suggests its potential usefulness as a wound healer.

Variations in the phenolic profile can be attributed to environmental factors (including planting conditions, soil, and location variability), extraction conditions, and the sensitivity of phenolic analyses.

### 3.2. Minimal Inhibition Concentration (MIC) Evaluation

Figure 2 and Figure 3 show the results of the minimum inhibitory concentration assay of the *Plantago major* extract against two different strains of *Staphylococcus aureus*, one susceptible (MSSA) and another methicillin-resistant (MRSA). *S. aureus* is a gram-positive opportunistic bacterium that colonizes the skin lesions, hampering wound-healing processes and increasing the severity of the lesions. It is often associated to nosocomial infections. Recently, the MRSA was indicated as one of the leading pathogens responsible for patients’ death related to antimicrobial resistance. In 2019, more than 100,000 deaths were attributed to this lineage [56].

For this test, a liquid medium was introduced in wells of a microplate and, through successive dilutions, the minimum concentration of the extract that inhibits bacterial growth after 48 h of culture was determined. The solution in the wells that contain the control group (culture medium without inoculum) remained clear after incubation, which indicates that there was no external contamination during the experiment. For both, the MSSA (Figure 2) and MRSA (Figure 3) strains, inhibition occurs for a concentration of 25% (*v*/*v*), which represents a concentration of 5.3 mg/mL of dry mass calculated from 10 mL of fresh extract, since these wells also present clear solutions. On the other hand, for lower concentrations, a turbid deposition was observed at the bottom of the microplate wells, indicating the proliferation of the bacteria in the medium [32]. For this reason, the PCL/*P. major* electrospun nanofibers was produced from solutions containing 5.3 mg/mL of the extract (MIC) and 10.6 mg/mL (2 × MIC).

### 3.3. Viscosity Evaluation

One of the main variables that influences the morphology of electrospun nanofibers is the viscosity of the solution to be spun. This is attributed to the ability of entanglement of the polymer chains that will allow their interaction during the spinning process [57]. The viscosity of PCL solution (10 wt. %) with different contents of *P. major* L. extract was performed at different shear rates (Figure 4). The studied solutions were PCL_0.0 (pure PCL 10 wt %), PCL_5.3 (PCL 10 wt % plus *P. major* at MIC concentration), and PCL_10.6 (PCL 10 wt % plus *P. major* at twice MIC concentration).

The three solutions show Newtonian behavior. As shown in Figure 4, the solution of PCL without extract (PCL_00), and that of PCL with 5.3 wt. % of extract (PCL_5.3) presented similar viscosity values throughout the range of shear rate tested. A slight viscosity reduction was observed for the solution with lower extract concentration (5.3 wt. %) in relation to that of the pure PCL solution, which may be related to a lubricant behavior of the molecules of the extract inside the polymeric network. This behavior was also observed in the results presented by Figueiredo et al. (2022) [58], who observed the lubricant behavior on viscosity when incorporating propolis extract into the PCL solution.

On the other hand, this behavior was not observed for the solution with the highest extract content (PCL_10.6), which showed an 18% increase in viscosity values as compared to the pure PCL solution. This result may be related to a possible supersaturation of the solution, creating an emulsion that affects the chemical interaction/affinity with the polymer; this behavior was described in the literature for polymeric solution with filler that presented different chemical affinities, forming an emulsion between filler/polymeric solution [59].

### 3.4. PCL/P. major Electrospun Samples

The PCL spinning study of this work started based on previous studies developed in our research group [42,58], with variations of solvent systems. The solution was prepared using PCL (10 wt. %) in DCM/DMF. Studies were performed to select the suitable experimental parameters to obtain fibrillar morphology without defects and with smaller and more homogeneous fiber diameters [42,44]. Based on this investigation, the following spinning conditions were chosen: flow rate = 0.75 m/h; voltage = 17 kV; needle tip–collector distance = 12 cm. Three solutions were spun using a 5 mL solution with different *P. major* extract concentrations, pure PCL (PCL_00, PCL_5.3, and PCL_10.6).

Figure 5A,B present the fibrillar morphology of neat PCL (233.2 ± 84.2 nm) and with 5.3 wt. % extract (235.5 ± 72.2 nm), respectively. It shows that adding this proportion of the extract in PCL did not significantly change the average diameter and fiber’s morphology, which was maintained with no significant presence of defects and beads for both samples. However, for the higher concentration of *P. major* in PCL (10.6 wt. %) (Figure 5C), a higher average diameter and a large distribution of diameters were observed in the fibers (378.5 ± 118.5 nm). These variations may be attributed to instabilities in the solution with the addition of higher amount of the extract, which could be related to the occurrence of a phase separation during the spinning processing due to the difference of polarity between PCL and extract or the supersaturation of the solution as suggested before. Analysis of variance (ANOVA) post hoc test–Fisher’s least significant difference (LSD) (calculated LSD = 0.877) indicates that increasing levels of *P. major* extract on the electrospun solution were statistically significant from 0 to 10 wt. % (LSD > 1.5 × 10^−12^), and from 5 to 10 wt. % (LSD > 1.5 × 10^−12^). However, the increment of 5 wt. % of vegetal extract concerning the pure PCL solution (LSD < 0.899) was not influential for the fiber diameter.

To evaluate the incorporation of the extract in the polymer matrix (PCL), the wettability test was performed on the surface of the electrospun mats. The image of the water drop on the surface of each mat was shown in Figure 5A–C. The results indicate a decrease in the contact angle of the mats formed by fibers containing the *P. major* extract, from 121.9° for neat PCL to 39.4° and 0° when the extract concentration is increased. The wettability is mainly related to the surface chemical interactions of the material as well as its morphology [47,55,58]. As shown in Figure 5, PCL_00 and PCL_5.3 samples presented fibrillar morphology with very similar fiber diameters, which mainly demonstrates that the variation of the contact angle for these samples may be related to the incorporation and functionalization of PCL nanofibers by the extract and the arrangement of the polar groups on the surface of the produced nanofibers [60]. The *P. major* extract has halogenated and OH-containing components, which makes it rich in hydrogen bonds, and with a large affinity to water, unlike PCL, which is a highly hydrophobic polymer. On the other hand, the PCL_10.6 sample, which showed total water absorption (CA = 0°), may be influenced by the two main phenomena, one related to the chemical character/composition of the extract, as well as, the capillarity of the mats, since a possible increase in the pores formed inside the fibers was verified. This feature generates a higher humectability capacity of their fibers compared to the other two samples evaluated [61,62]. It is well known that a material with hydrophilic character may provide conditions for cell attachment, proliferation, migration, and differentiation.

### 3.5. Fourier Transformation Infrared—FTIR

Figure 6 presents the FTIR-ATR spectra of PCL electrospun mats and the freeze-dried *P. major* extract following reports in the literature [63]; it was easy to identify from the PCL spectrum that the band attributed to carbonyl group stretching (C=O) at approximately 1750 cm^−1^ [47]. Absorption bands related to the solvents used in the electrospinning process were not found; thus it can be stated that, if present, the solvents are in very low quantity in the fibers having been evaporated almost completely during the spinning process.

The spectrum of the extract shows a broad band at 3275 cm^−1^, which is characteristic of the vibrations of phenolic groups and O-H bonds of polyphenols and flavonoids. The presence of a band at 1719 cm^−1^ due to C=O bond also present in polyphenols and flavonoids of the *P. major* extract was also observed [38,64,65]. A characteristic band of *P. major* at 1558 cm^−1^ was observed in the FTIR spectra of PCL_5.3 and PCL_10.6 samples, demonstrating the incorporation of the *P. major* extract in the polymer nanofibers [66,67,68]. The incorporation of the extract in the fibers was corroborated since the transmittance band of the carbonyl characteristic of PCL (C=O) at 1719 cm^−1^ had a slight shift to higher values of 1723 cm^−1^ for PCL_5.3 and 1726 cm^−1^ for the sample with 10.6 wt. % (PCL_10.6), which can be attributed to the formation of intermolecular hydrogen bonds between PCL and the *P. major* extract [40,69,70].

### 3.6. Thermal Behavior of Electrospun Mats

The thermal behavior for the three samples evaluated in this study was characterized using DSC (Figure 7) and TGA with the aim of evaluating the influence of the *P. major* extract on the thermal transitions and thermal stability of PCL for each sample. The values of melting temperature (T_m_), enthalpy of melting (ΔH_m_), and degree of crystallinity (Xc) are shown in Table 1.

The electrospinning process generates rapid elongation of polymer chains that allow their alignment, promoting high structural organization [70] and degree of crystallinity than that measured in the second heating cycle as shown in Figure 7A. However, this process can regenerate variations in the population of crystalline lamellae that are evidenced in the first heating cycle as two main types of crystals: one less perfect melting at 49 °C approximately and a second population (in larger proportion) evidenced by a higher ΔH_m_ at 55.1 °C. The same behavior was seen for samples with two different natural extract contents.

For the second heating cycle, Figure 7B, the PCL electrospun mats with the *P. major* extract showed a decrease of approximately 36.6% in the degree of crystallinity of the fibers with 5.3 wt. % of the extract compared with PCL fibers without the extract (PCL_00) (Table 1). This demonstrates an interaction between freeze-dried extract and PCL chains that can be attributed to the fact that the *P. major* molecules were able to allocate themselves between polymer chains, promoting the increase in the space between them and increase in the difficulty of these chains in reorganizing as ordered crystal structures [58,71]. Nevertheless, this large decrease in the degree of crystallinity was not observed when the concentration of the extract was increased to 10.6 wt. % (PCL_10.6), and a slight increase in crystallinity was observed in relation to PCL_5.3. This can be attributed to a possible supersaturation of the extract solution which creates phase separation in the solution, resulting in a low homogeneity or dispersion of the extract in the electrospun nanofibers. Although PCL_10.6 has lower Xc compared to the neat PCL fibers (PCL_00), it has Xc about 16% higher than sample PCL_5.3.

Figure 7C,D shows TG and DTG curves of the PCL/*P. major* electrospun mats, which allow to infer about the thermal stability of the samples; the thermal events for the samples and extract are referred in Table 2. Three stages of mass loss are evidenced for the *Plantago major* extract (Table 2). The first stage occurs between approximately 45–167 °C and is attributed to loss of the water present in the sample [42]. This stage was not observed in any electrospun sample, demonstrating the absence of small molecules like water and solvent in the nanofibers (Figure 7C). However, three stages of mass loss at higher temperatures were seen in the PCL/*P. major* electrospun mats, two of them with maximum DTG mass loss rate (peak maximum) at 181 and 256 °C, attributed to components of extract, and one between 320 and 450 °C related to the stage of PCL chain degradation [58,72].

### 3.7. In Vitro Swelling Behavior of Electrospun Nanofibers

Keeping a moist environment in the wound is of great importance to promote the different stages of tissue regeneration, favoring the formation of granular tissue, thus accelerating re-epithelization in the first 48 h after dressing application [73]. Another important feature in the implementation of dressing for skin, besides keeping this moisture, is the non-retention of the absorbed exudate and fluids excessively, creating maceration in the tissue surrounding the wound [74].

Figure 8 shows the degree of swelling and morphology of the electrospun mats evaluated in this study. PCL/*P. major* electrospun mats were immersed in PBS (pH 7.4) for 96 h. Subsequently, the samples were frozen using liquid nitrogen, freeze-dried at 24 h using lyophilizate equipment, and, finally, analyzed using SEM (Figure 8c–e). The comparative diameters of fibers before and after the swelling/release assay and the mass variation of mats are shown in Table 3.

Electrospun mats have a high contact area and porosity attributed to their fibrillar morphology. This allows the surrounding medium (PBS) to penetrate the mat due to capillary effects, independent of the hydrophobic character of PCL as observed in the contact angle evaluation (Figure 8). The pure PCL fibers (PCL_00) showed a PBS absorption capacity of ~22% for the first hours (Figure 8B), as well as a constant increment during the test up to approximately 200% a result that agrees with the results shown in the literature for electrospun pure PCL fibers [24]. On the other hand, this behavior was not observed for the samples with the *P. major* extract. It was shown to have a higher swelling capacity. This can be attributed to the location of the extract on the surface of the fibers allowing better interaction mat/PBS.

Subsequent to the swelling test, the samples were weighed and the morphology obtained from SEM images compared with the initial nanofibers. The results of mass variation and fiber’s morphology (Table 3 and Figure 8c–e) suggested that no degradation of the electrospun PCL fibers by the PBS medium took place for both materials, with and without the extract. Although a low mass variation was observed for PCL_00 mat after the swelling test in PBS, it seems that small quantity of salts present in the liquid medium was absorbed by the hydrophobic PCL.

On the other hand, the fiber diameter showed significant variation after the swelling test (freeze-dried), even for the neat PCL fibers. The fibers showed diameter increase of approximately 10.7%, 11.2%, and 13.1% for PCL_00, PCL_5.3, and PCL_10.6, respectively, demonstrating the absorption capacity after 96 h submerged in PBS. The result is in concordance with reports made in the literature for PCL/propolis electrospun mats, which demonstrates the ability to maintain the moistness of the wound for its recovery [47,58].

### 3.8. P. major Extract Encapsulation

The capacity of encapsulation of the *P. major* extract in PCL by electrospinning was evaluated using UV-Vis spectroscopy using a thin film accessory. It was evaluated using two methods: the drug-loading capacity (DL), which is the amount of drug loaded per unit weight of each mat, and the encapsulation efficiency (EE), which describes the concentration of the active material incorporated on electrospun mats and the relationship with the initial *P. major* extract amount placed into the polymeric solution [48]. The result is shown in Table 4.

We found that the experimental DL of *P. major* in nanofiber’s mats was approximately 50% of what would be expected theoretically. Similar behavior was observed for experimental EE, where it showed minor values (~40%) compared with the theorical value. This result can be attributed to the chemical characteristic differences between the PCL-*P. major* extract generating some instability of extract in the polymeric solution as mentioned previously. This was corroborated with the variation of the amount of *P. major* extract in solution, increasing the instability in the polymeric solution and forming fibers with major variation in the extract concentration. This results in decrease in the encapsulation amount as shown in Table 4, for EE.

### 3.9. Water Vapor Permeation Test

The water vapor permeation capacity of dressings is a key measure when they are intended for use in wound care. This is because of the need for wound breathability and the exchange of vapor with the environment, which helps to maintain moisture. Dressings that allow for a very high rate of transpiration may promote wound dryness, thereby preventing an effective environment for cellular migration and proliferation. To address this, using Fick’s law, the WVRT was calculated for the samples studied, and the results are reported in Table 5.

Table 5 shows the WVTR values of electrospun PCL/*P. major* samples compared to a control measurement (empty cup). The PCL_5.3 matrix had a higher transpiration rate (1545.98), followed by the PCL mats with 10.6% extract (PCL_10.6) and pure PCL mats (PCL_00). This behavior was expected, as the PCL_5.3 samples showed a higher liquid adsorption capacity (swelling), demonstrating that the swelling capacity is directly related to the transpiration rate, promoting better exchange of water vapor with the environment, thus allowing for greater breathability of the wound. On the other hand, the WVTR values of commercial fibrous dressings have a range of 1263–2179.84 g/m^2^ per day, mainly attributed to their porosity [56,75]. Finally, all matrices produced in this work showed permeation values within the commercial range, demonstrating their potential for application as a skin dressing.

### 3.10. P. major Extract Release Assay

To evaluate the delivery of the encapsulated extract, release assay of electrospun PCL nanofibers was performed in PBS (pH 7.4) at 35 °C under stirring (100 rpm). Aliquots were taken at different times up to 48 h. The release was quantified using UV-Vis spectrophotometer. Quantifications of the *P. major* extract release were evaluated for the two different *P. major* concentrations, 5.3 and 10.6 wt. % (Figure 9).

The highest speed of extract release was achieved in the first 24 h of assay, which is a significant result considering the use of these materials in wound dressing, since clinically the time of using a dressing may not exceed 24 h. This occurs because the adsorption of exudate generates bad odors and discomfort to patient and health staff.

The delivery profile from monolithic nanofibers showed more controlled release, since the burst effect characteristic of electrospun nanofibers was not observed. The maximum delivery rate of the extract was observed after the first 24 h of assay for both concentrations of the encapsulated extract [70,76].

Figure 9 also presents the cumulative delivery of the *P. major* extract in PBS, which shows deliveries of 0.17 mg/mL and 0.23 mg/mL in the first minutes of the assay (30 min) and a total delivery of 0.35 mg/mL and 0.56 mg/mL for PCL_5.3 and PCL_10.6, respectively, after 48 h assay. These values are within the values recorded for bacterial inhibition reported in the literature (Grand-negative, and fungus) for studies using *P. major* extracts from leaves (between 0.1 and 1.0 mg/mL) [22,77].

### 3.11. Antimicribial Assay from PCL/P. major Electrosun Nanofibers

One of the most relevant properties cited in the literature for *P. major* is the content of flavonoid, tannins, phenolics compounds among other substances that contribute to its antimicrobial action for leaves and stem, as well as for the seeds in the preparation of the mucilage [78].

Figure 10 shows the results of antimicrobial activity of the *P. major* extract and PCL_5.3 electrospun mats. As a positive control, samples of neat PCL (PCL_00) mats were used. For the negative control, the lyophilized extract on filter paper and `dry was used.

This study demonstrated the capacity of the *P. major* extract to eliminate *E. coli*, as shown in Figure 10. The figure also shows that the action of the pure extract against *S. aureus* was not so significant with only a slight decrease in the bacterial count when compared to the positive control. The low antimicrobial activity of the pure extract may be related to its low content of total flavonoids, as discussed before. This result agrees with the literature, which reports that the hydroalcoholic extract shows the highest inhibition for gram-negative bacteria [79,80]. The antimicrobial assay showed, however, that for the PCL/*P. major* sample the decrease in proliferation of both *E. coli* and *S. aureus* bacteria strains was approximately 60%. Complete inhibition was not observed for the used strains, a result that agrees with those recorded in the literature [81,82]. On the other hand, this partial inhibition can be explained considering possible interactions between the extract and the polymer that did not allow the complete release of the extract, as shown in Figure 9, avoiding the achievement of the minimum inhibitory concentration of these strains. Thus, other studies are necessary to evaluate the release of other concentrations of the extract that can inhibit these bacteria strains.

## 4. Conclusions

This work studied the preparation and characterization of electrospun PCL mats with the incorporation of *Plantago major* L. extract as an active material or phytopharmaceutical. The use of phytopharmaceuticals or natural products in the formulation of new materials for biomedical applications has been growing in recent years. *P. major* is a widely used plant worldwide for medical applications with satisfactory results attributed to the components present in the extract, which have anti-inflammatory, antifungal, analgesic, antioxidant, and other properties. In this work, it was possible to observe the presence of secondary metabolites such as hydroxycinnamic acid, chlorogenic acid, Rosmarinus acid, and luteolin, a flavonoid, components evaluated using HPLC, and components related to the antioxidant and antibacterial behavior of *P. major*. This was corroborated in this study by achieving a reduction in the proliferation of methicillin-sensitive and resistant S. aureus and *E. coli*, demonstrating its potential for use in wound treatment. On the other hand, this study demonstrated the encapsulation capacity of the extract in nanofibers produced by PCL electrospinning, potentially reducing the possible degradation of the extract when used directly on the wound. It was also observed that the PCL/*P. major* electrospun matrices presented effective conditions for use as dressings due to their exudate absorption capacity (swelling degree), maintaining a moist environment favorable for cellular migration and proliferation with wound transpiration capacity compared to WVPT values of dressings available on the market, and preventing the penetration of environmental agents and/or pathogens into the wound, leading to biofilm formation. Based on this, it can be concluded that the material produced in this work with a concentration of 5.3 wt. % of the *P. major* extract (PCL_5.3) offers great potential for use as an advanced dressing in the treatment of chronic wounds with difficult healing, thus encouraging future in vitro and in vivo biological tests to evaluate the proliferation and cellular activity of keratinocytes and fibroblasts.

## Figures and Tables

**Figure 1 pharmaceutics-15-01047-f001:**
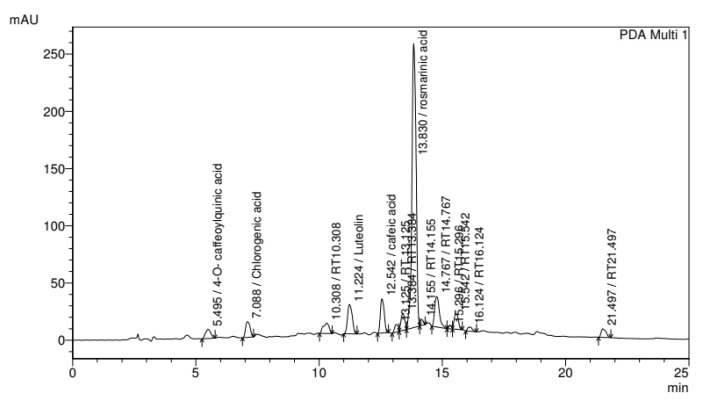
HPLC chromatogram from *P. major* extract obtained by C-18 column (250 mm × 4.6 mm × 5 µm), flow rate 1.0 mL/min, and mobile phase composed by water: acetic acid (99:1, solvent A) and methanol (solvent B) at 320 nm.

**Figure 2 pharmaceutics-15-01047-f002:**
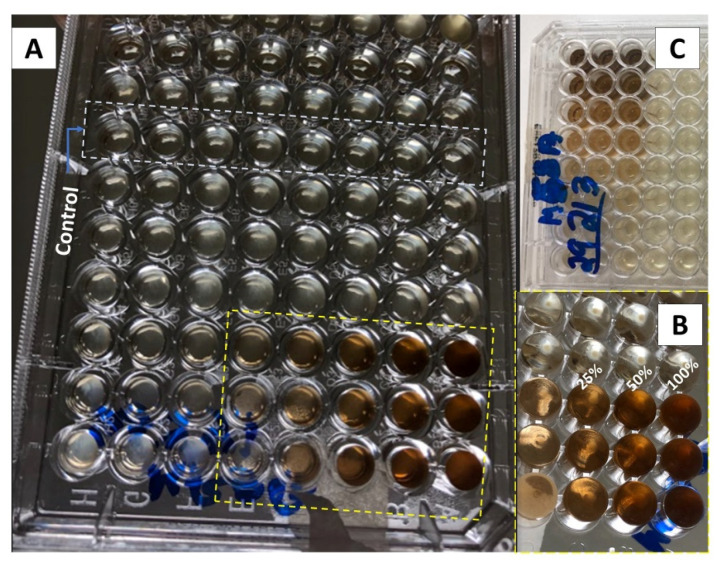
Inhibitory dose study in microplates with dilution for MSSA *S. aureus*. (**A**) Photograph of the bottom of the microplate highlighting the wells with the control group and the 48 wells with the extract using the methicillin-susceptible *S. aureus* strain (MSSA). (**B**) Enlarged photograph for the wells with 100, 50, and 25% of the extract. (**C**) Photograph of the top of the plate with the bacterial strain (MSSA).

**Figure 3 pharmaceutics-15-01047-f003:**
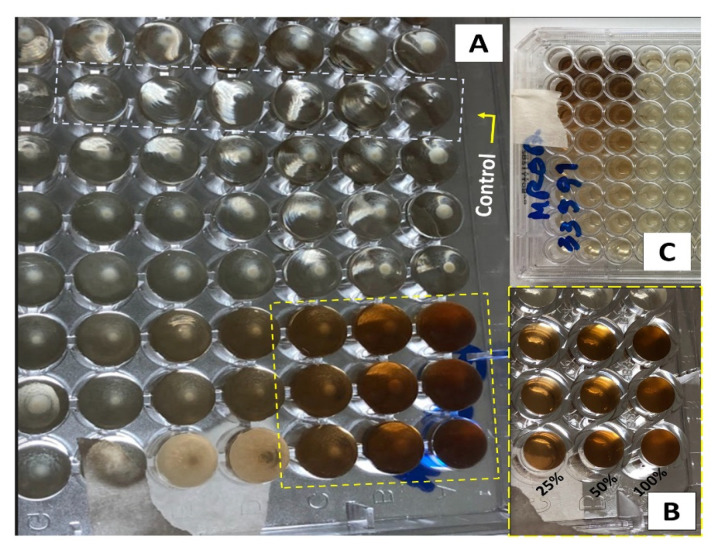
Microplate inhibitory dose study for resistant *S. aureus* (MRSA). (**A**) Photograph of the bottom of the microplates highlighting the wells with the control group and the 48 wells with the extract. (**B**) Enlarged photograph for the wells with 100, 50, and 25% of the extract. (**C**) Photograph of the upper part of the plate with the type of bacteria (MSRA).

**Figure 4 pharmaceutics-15-01047-f004:**
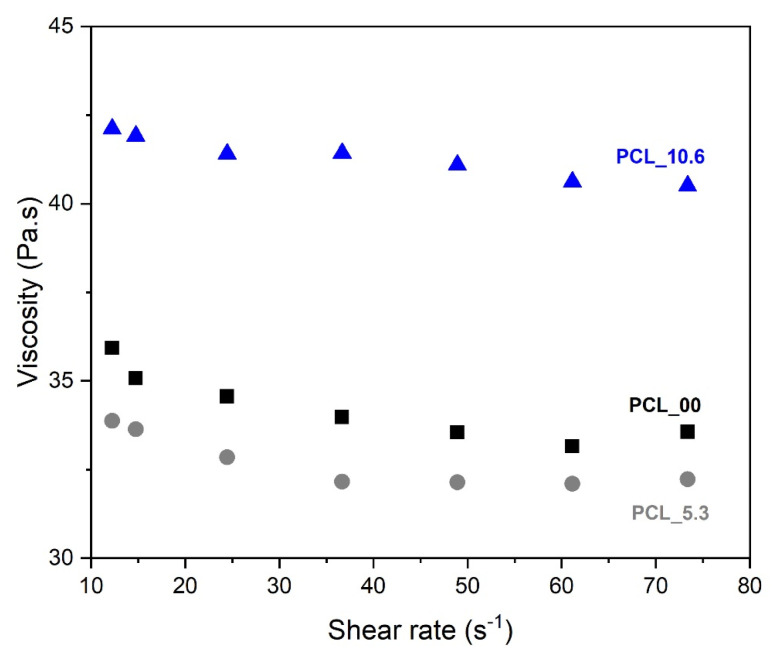
Rheological behavior of PCL solutions with different contents of *Plantago major* dry extract.

**Figure 5 pharmaceutics-15-01047-f005:**
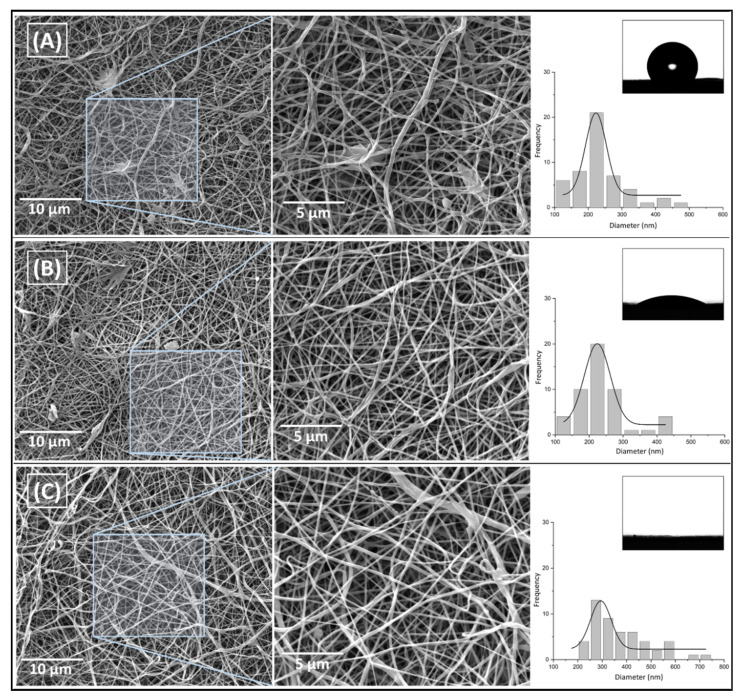
SEM images of electrospun mats with magnification variation and their histograms at varying concentrations of *P. major* extract in the relationship of PCL and the image of water drop on the surface of the mat that was taken during contact angle (CA) analysis. (**A**) PCL nanofibers without extract (PCL_00%); (**B**) PCL nanofibers with 5.3% of the extract (PCL_5.3%), and (**C**) PCL nanofibers this 10.6% of the extract (PCL_10.6%).

**Figure 6 pharmaceutics-15-01047-f006:**
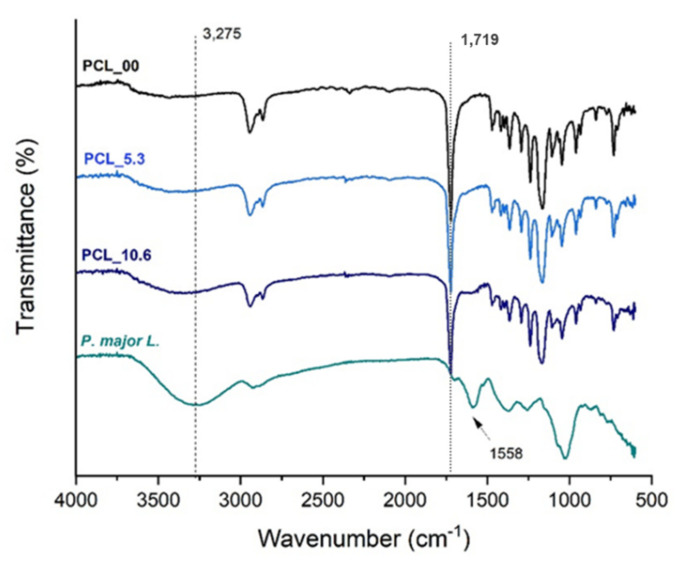
FTIR-ATR spectra of the PCL electrospun mats with different amounts of *Plantago major* extract and the freeze-dried extract.

**Figure 7 pharmaceutics-15-01047-f007:**
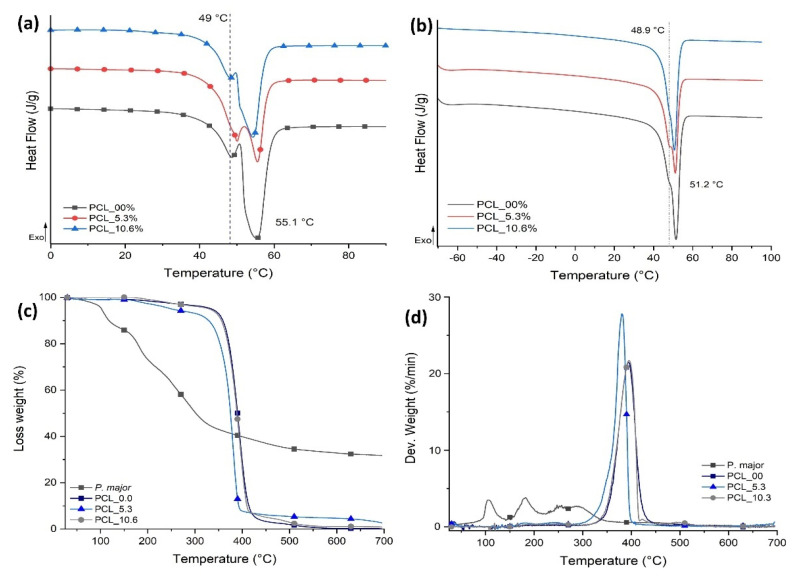
Thermal behavior PCL/*P. major* electrospun samples: DSC curves for electrospun mats from PCL with different concentrations of *P. major* extract. (**a**) First heat cycle and (**b**) Second heat cycle; TG (**c**) and DTG (**d**) curves.

**Figure 8 pharmaceutics-15-01047-f008:**
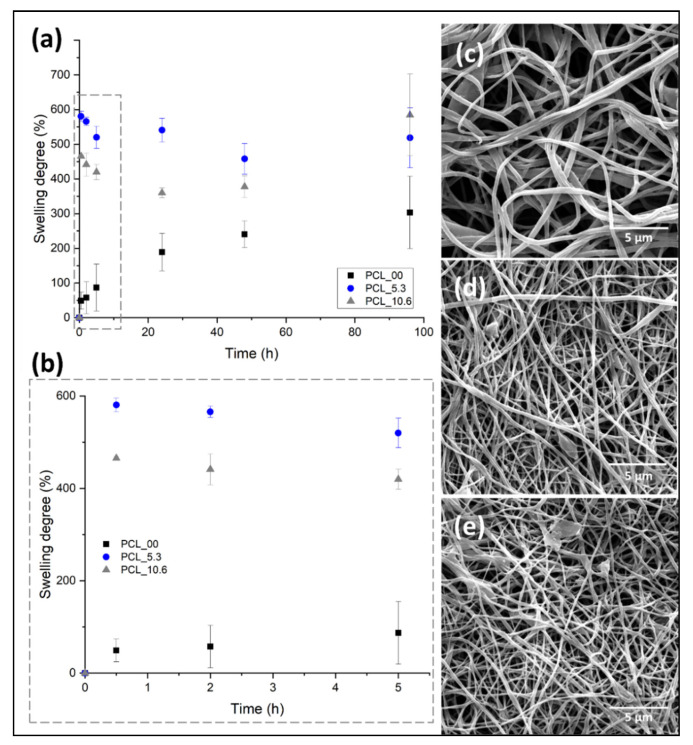
Degree of swelling for PCL/*P. major* electrospun mats as a function of the immersion time in phosphate buffer solution—PBS (pH: 7.4): (**a**) 96 h of the assay, (**b**) first 5 h of swelling assay; SEM images of *PCL/P. major* electrospun mats after swelling assay (96 h): (**c**) PCL_10.6; (**d**) PCL_5.3, and (**e**) PCL_0.0.

**Figure 9 pharmaceutics-15-01047-f009:**
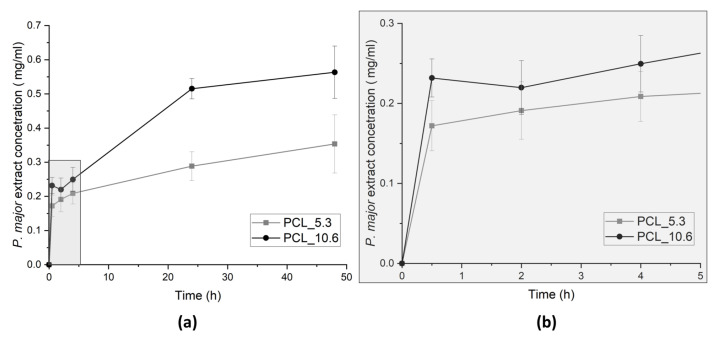
*P. major* extract release from PCL/*P. major* electrospun mats as a function of the immersion time in phosphate buffer solution—PBS (pH 7.4): (**a**) 48 h of assay, and (**b**) zoom of the evolution of the release assay in the first 5 h.

**Figure 10 pharmaceutics-15-01047-f010:**
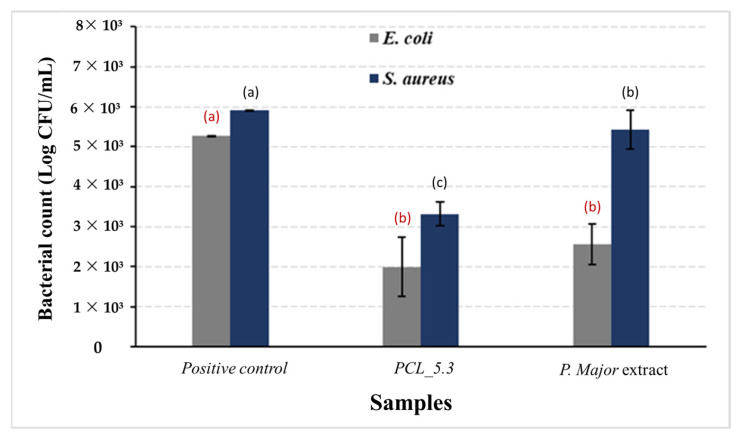
Antimicrobial activity of *Plantago major* hydroalcoholic extract and PCL/*P. major* electrospun mat against *E. coli* and *S. aureus* strains. Pure PCL electrospun mat (PCL_0.0) was used as positive control (a,b,c Different letters indicate statistically differences (Kruskal–Wallis and Fisher’s test; *p* < 0.05).

**Table 1 pharmaceutics-15-01047-t001:** Melting temperature and crystallinity degree of electrospun mats of PCL/*P. major*.

Sample	1st Heating Cycle			2nd Heating Cycle		
	*T_m_* (°C)	Δ*H_m_* (J/g)	*X_c_*	*T_m_* (°C)	Δ*H_m_* (J/g)	*X_c_*
PCL_0.0	55.4	123.0	81.1	51.4	117.0	77.1
PCL_5.3	55.4	80.5	53.1	51.1	74.2	48.9
PCL_10.6	54.3	88.5	58.3	50.6	86.0	56.7

**Table 2 pharmaceutics-15-01047-t002:** Thermal characteristics (TGA) of electrospun mats of *P. major*, and PCL with different compositions of extract.

Degradation Event		T_onset_ (°C)	T_max_ (°C)	T_endset_ (°C)	Mass Loss (%)
*P. major*	1°	45	103.0	167.1	11.9
	2°	167.1	182.3	239.6	18.1
	3°	239.7	287.1	438.8	38.2
PCL_00	1°	368.7	394.0	445.3	99.9
PCL_5.3	1°	170.0	180.6	237.2	04.8
	2°	360.2	380.7	430.4	92.5
PCL_10.6	1°	170.7	192.4	367.9	05.0
	2°	367.9	394.6	457.5	94.0

**Table 3 pharmaceutics-15-01047-t003:** Comparative diameters and mass variation after assay of the electrospun nanofibers before and after swelling with lyophilized electrospun mats.

Sample	Fiber Diameters (nm)		Mass Variation
	Before	After	(%)
PCL_00	233.2 ± 84.2	258.3 ± 77.8	3.7 ± 0.38
PCL_5.3	235.5 ± 72.2	261.9 ± 84.4	7.1 ± 0.15
PCL_10.6	378.5 ± 118.5	428.1 ± 136.9	30.4 ± 10.2

**Table 4 pharmaceutics-15-01047-t004:** *Plantago major* loading capacity and encapsulation efficiency in PCL nanofibers (n = 3).

Sample	Drug-Loading Capacity (DL)		Encapsulation Efficiency (EE)	
	Theoretical (%)	Experimental (%)	Theoretical (%)	Experimental (%)
PCL_5.3	25	13.02	~100	63.52
PCL_10.6	50	31.01	~100	60.31

**Table 5 pharmaceutics-15-01047-t005:** The water vapor transpiration rates (WVTR) of PCL/*P. major* mats.

Sample	J = Δm/Δt (g/min)	Water Vapor Transpiration Rate (g/m^2^ per day)
PCL_00	0.01008 ± 0.00083	1471.34
PCL_5.3	0.01213 ± 0.00047	1545.98
PCL_10.6	0.01122 ± 0.00045	1492.83

## Data Availability

The data that support the findings of this study are available from the corresponding authors upon reasonable request.

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
