# Peer review of "Electrospun Nanofibers Loaded with Plantago major L. Extract for Potential Use in Cutaneous Wound Healing"

_pharmaceutics, 2023, doi:10.3390/pharmaceutics15041047_

Round 1

Reviewer 1 Report

Anaya-Mancipe et al. fabricated Plantago major L. extract-loaded PCL electrospun nanofiber mats, and explored their performances for wound healing application. Some major revisions should be conducted before publication.

1. Please state the reasons why Plantago major L. extract was chosen in this study. As well known, lot of different plant extracts have been applied for wound healing application in the recent years. What are the merits and advantages of Plantago major L. extract compared with some other biological plant extracts?

2. The advantages and disadvantages of PCL should also be given, compared with some other biopolymers like PLLA, and PLGA, etc. The PCL polymer usually spends nearly two years for degradation in vivo. Is it suitable for the fabrication of wound dressings?

3. The merits of electrospinning is suggested to be outlined in the introduction section. Some recent works about the innovative electrospinning like European Polymer Journal, 2023, 186, 111863, https://doi.org/10.1016/j.eurpolymj.2023.111863 and ACS Applied Materials & Interfaces, 2022, 14(14), 15911-15926, https://doi.org/10.1021/acsami.1c24131 are suggested to be discussed.

4. Please state the reasons for the selection of polymeric and extract concentration as well as solvent ratio for electrospinning. Moreover, how did the authors choose the parameters of electrospinning? Do they conduct any preliminary experiments?

5. Please explain why the viscosity of pcl_5.3 solution was lower than PCL solution, but the viscosity of pcl_10.6 solution was higher than PCL solution in Figure 4.

6. The as-related thermal parameters for TG analysis are suggested to be presented in the form of a table, in order to increase the readability.

7. In Figure 8, please justify the reasons why the PCL_5.3 exhibited the highest swelling degree.

8. How did the authors observe the fibers after swelling in Figure 8? And the related descriptions should be given in a much detailed way.

9. Did it have any significant difference in Figure 10? The statistical analysis method should be conducted.

10. The grammar and writing should be improved in the whole manuscript.

Author Response

Dear Reviewer,

Thank you again for the careful review of the manuscript. We want to acknowledge the constructive comments and suggestions. For convenience, implemented changes have been highlighted in red in the revised manuscript. Please find below the point-by-point responses to all comments. 

With kind regards,

On behalf of my co-authors,

Prof. Rossana Mara da Silva Moreira Thiré

Prof. Marcos Lopes Dias

Reviewer 2 Report

The paper deals with an interesting subject, which is the design and characterization of polycaprolactone (PCL) loaded with Plantago Major L. extract for use in cutaneous wound healing by electrospinning. The paper is of clear interest for the reader. It is well written and referenced. Nevertheless, the paper is not suitable for publication in the present form and some major changes should be properly addressed by the authors to improve the clarity and the understanding.

I would suggest the authors to divide the materials and methods sections in different paragraphs: physico-chemical characterization of the material and patch, in vitro release of the extract, in vitro microbiological testing and in vitro cell culture.

In the title, the authors talk about wound healing. However, there are no experiments related to the use of wound related cells: human dermal fibroblasts, keratinocytes, huvecs and so on. I would suggest to test the biocompatibility, and perform a scratch assay to see the effect of the material on cell migration.

I would suggest replacing figure 2 and 3 with more clear pictures.

The authors should improve the quality of Figure 7.

Statistical Analysis is missing.

In the conclusion section, the authors should be better highlight the results achieved.

Author Response

Dear Reviewer,

Thank you again for the carefully review of the manuscript. We would like to acknowledge the constructive comments and suggestions. For convenience, implemented changes have been highlighted in red font in the revised manuscript. Please find below point-by-point responses to all comments. 

With kind regards,

On behalf of my co-authors,

Prof. Rossana Mara da Silva Moreira Thiré

Prof. Marcos Lopes Dias

Round 2

Reviewer 1 Report

The reviewer's comments have been addressed well.

Author Response

The authors thank you for the review and consideration made for this manuscript.

Reviewer 2 Report

Dear Editor,

The paper can be accepted in the present form. Even if the authors did not address all the issues, they improved the quality and the understanding of the paper.

Best regards

Author Response

(The authors gave the same response as above.)
